# Decoration of Polyfluorene-Wrapped Carbon Nanotubes with Photocleavable Side-Chains

**DOI:** 10.3390/molecules28031471

**Published:** 2023-02-03

**Authors:** Dialia Ritaine, Alex Adronov

**Affiliations:** Department of Chemistry and Chemical Biology, Brockhouse Institute for Materials Research, McMaster University, 1280 Main Street W, Hamilton, ON L8S 4M1, Canada

**Keywords:** carbon nanotubes, conjugated polymers, photocleavable sidechains, supramolecular functionalization, click chemistry

## Abstract

Functionalizing polyfluorene-wrapped carbon nanotubes without damaging their properties is effective via Copper-Catalyzed Azide–Alkyne Cycloaddition (CuAAC). However, the length and nature of polymer side-chains can impact the conductivity of polyfluorene-SWNT films by preventing close contact between the nanotubes. Here, we investigate the functionalization of a polyfluorene-SWNT complex using photocleavable side-chains that can be removed post-processing. The cleavage of the side-chains containing an ortho-nitrobenzyl ether derivative is efficient when exposed to a UV lamp at 365 nm. The photoisomerization of the *o*-nitrobenzyl ether linker into the corresponding *o*-nitrosobenzaldehyde was first monitored via UV-Vis absorption spectroscopy and ^1^H-NMR spectroscopy on the polymer, which showed efficient cleavage after 2 h. We next investigated the cleavage on the polyfluorene-SWNT complex via UV-Vis-NIR absorption spectroscopy. The precipitation of the nanotube dispersion and the broad absorption peaks after overnight irradiation also indicated effective cleavage. In addition, Raman spectroscopy post-irradiation showed that the nanotubes were not damaged upon irradiation. This paper reports a proof of concept that may find applications for SWNT-based materials in which side-chain removal could lead to higher device performance.

## 1. Introduction

Since the discovery of single-walled carbon nanotubes in 1991 [1], significant effort has been made to take advantage of their mechanical [2,3], optical [4,5], and electronic properties [6,7]. Numerous applications of SWNTs such as sensors [8,9], thin-film transistors [10,11], organic photovoltaics [12,13], flexible electronics [14,15], and conductive inks [16,17] have been developed [18,19]. However, due to inter-tube π–π interactions, SWNTs tend to aggregate into insoluble bundles in organic solvents [20]. Moreover, all commercial techniques to produce SWNTs [21,22] result in a complex mixture of amorphous carbon, leftover catalyst particles, and a mixture of semiconducting and metallic species (sc- and m-SWNTs, respectively) that impede their performance within several applications [23]. To address this issue, covalent or non-covalent functionalization of SWNTs has been developed to improve their solubility and purity. Covalent functionalization requires strongly oxidizing conditions that damage SWNTs’ properties by destroying the sp^2^ hybridized surface and therefore impact on SWNTs’ properties [24,25]. In contrast, non-covalent functionalization uses sonication in the presence of a dispersant to form a dispersant-SWNTs supramolecular complex that provides solubility in organic solvents and prevents SWNTs reaggregation into bundles [26,27,28]. To this end, dispersants such as small aromatic compounds [29,30,31], surfactants [32,33,34], biomolecules [35,36,37], and conjugated polymers [38,39,40,41] have been used. Due to their facile synthesis and structural modification [42,43,44,45,46], conjugated polymers can be used to achieve different properties such as selective dispersion of either sc- or m-SWNTs [47,48,49,50], reversible assembly on the nanotube surface [51,52,53,54,55,56], or depolymerization in response to a stimulus to release SWNTs [57,58,59].

Attention has also been given to conjugated polymers that exhibit reactive functionality post-supramolecular assembly with SWNTs. Specifically, polyfluorene derivatives containing azide groups in the side-chains have been prepared and used to noncovalently functionalize SWNTs [60,61,62]. The resulting polyfluorene-SWNTs complexes, dispersed in organic or aqueous solvents, were then functionalized using either Copper-Catalyzed Azide–Alkyne Cycloaddition (CuAAC) or Strain-Promoted Azide–Alkyne Cycloaddition (SPAAC) without damaging SWNTs’ optoelectronic properties [60,61,62]. However, the introduction of large side-chains that are non-conductive results in a dramatic decrease in conductivity of the polymer-SWNTs complex by preventing good contact between adjacent nanotubes. A potential approach to solve this problem is the use of cleavable side-chains that can be removed after the device’s fabrication. In 2019, Kawamoto and coworkers prepared SWNTs thin films dispersed using polythiophene functionalized with carbonate linkers in their side-chains. When heated at 350 °C, the carbonate linkers were cleaved by decarboxylation resulting in higher conductivity [63]. More recently, our group prepared a polyfluorene-SWNTs complex functionalized with thermally cleavable side-chains that also contain a carbonate linker. The conductivity increased over time upon heating the films at 170 °C and reached a plateau of (2.0 ± 0.1) × 10^−2^ S/m after 17 h of heating, which was 20 times higher than the non-cleavable sample. This demonstrates the effect of removing the side-chains post processing [64].

Here, we report a different type of cleavable side-chain that contains an ortho-nitrobenzyl (*o*NB) ether linker. This linker is cleaved when exposed to a 300–365 nm UV light [65,66]. The cleavage time varies from minutes to a few hours, depending on the intensity of the light, which usually ranges from 1 to 60 mW.cm^−2^ [65]. *o*NB linkers have been widely used as cross-linkers for photodegradable polymers [67], side-chain functionalization [68,69,70,71,72], solid-phase synthesis [73,74,75], photolithography [76,77], self-immolative polymers [78,79], and other applications [65,66,80,81]. In this work, we prepared a polyfluorene-SWNTs complex functionalized with either photocleavable or non-cleavable side-chains. The cleavage study was first performed on the polymers using UV-Vis absorption and ^1^H NMR spectroscopy. We next studied the polyfluorene-SWNTs complexes using UV-Vis-NIR absorption spectroscopy. Characterization post irradiation was also performed using Raman spectroscopy to verify that the irradiation did not damage the nanotubes.

## 2. Results and Discussion

### 2.1. Polymer Synthesis and Characterization

To begin our study, we first synthesized an azide-containing polyfluorene (**PF-N_3_**) according to procedures in the literature [62]. Bromination of fluorene using *N*-bromosuccinimide (NBS) was performed to obtain precursor **1** (Appendix A), which was then alkylated with 1,6-dibromohexane to afford monomer **2**. Borylation of this monomer using Miyaura conditions afforded the diboronic ester **3.** Monomers **2** and **3** were then copolymerized using a Suzuki polycondensation to obtain the homopolymer (**PF-Br**) (Figure 1). Gel permeation chromatography (GPC) revealed an M_n_ of 33 kDa and a dispersity (Đ) of 2.2. **PF-N_3_** was then prepared via reaction between **PF-Br** and NaN_3_ in the presence of *^n^*Bu_4_NBr. The homopolymers were characterized by ^1^H NMR spectroscopy to confirm the presence of alkyl azides (3.15 ppm) in **PF-N_3_** and the disappearance of the signal of the alkyl bromides (3.31 ppm) in **PF-Br** (Appendix A).

To introduce our photocleavable side-chains, we first prepared an ***o*NB-TEG-alkyne** via activation of triethyelene glycol monomethyl ether with tosyl chloride (Appendix A), followed by alkylation with 5-Hydroxy-2-nitrobenzaldehyde to obtain compound **5**. Compound **5** was then reduced using sodium borohydride to afford the corresponding alcohol **6**. Nucleophilic substitution of this alcohol with propargyl bromide was finally performed to obtain ***o*NB-TEG-alkyne**. The non-cleavable analog was prepared via nucleophilic substitution of triethylene glycol monomethyl ether (TEG-OH) with propargyl bromide to give **TEG-alkyne** (Appendix A). **PF-N_3_** was then functionalized with either **TEG-alkyne** (**P1**) or ***o*NB-TEG-alkyne** (**P2**) (Figure 1) using copper-catalyzed azide-alkyne cycloaddition (CuAAC) (see Appendix A for details). The reaction was monitored by infrared (IR) spectroscopy via the disappearance of the polymer azide stretch at ~2090 cm^−1^ (Appendix A). The resulting polymers were also characterized by ^1^H–NMR spectroscopy to confirm the disappearance of the alkyl azides (3.15 ppm), the appearance of the aromatic proton in the triazole ring (7.51 ppm), as well as the appearance of alkyl protons from the side-chain between 3 and 5 ppm (Appendix A).

### 2.2. Photocleavage Study of The Polymers

With **PF-TEG** (**P1**) and **PF-oNB-TEG** (**P2**) in hand, we studied the cleavage of the side-chain using UV-Vis absorption spectroscopy. To confirm that the polyfluorene does not degrade upon irradiation, **PF-N_3_** was also studied. Polymers were dissolved in tetrahydrofuran (THF), transferred into quartz cuvettes, and irradiated at 365 nm in a UV reactor. The UV-Vis spectrum of each sample was measured every 15 min for a total duration of 2 h, when no further changes in absorbance were observed (Figure 2). Not surprisingly, **PF-N_3_** and **PF-TEG** do not show any degradation during the irradiation process (Figure 2A,B). As shown in Figure 2C, when **P2** was subjected to the irradiation process, a decrease in the absorbance is observed at 300 nm, corresponding to the cleavage of the *o*-nitrobenzyl linker. Meanwhile, as the peak at 300 nm decreases, a new peak at 350 nm arises. This new peak corresponds to the *o*-nitrosobenzaldehyde compound produced during the cleavage (see Appendix A) [65]. As shown in the spectrum, this new peak overlaps with the absorption of the polyfluorene. Therefore, the cleavage was also monitored on the ***o*NB-TEG-alkyne** side-chain (Figure 2D) and provided similar observations. This suggests that the cleavage is effective, and 2 h is sufficient to cleave the side-chain.

To further characterize the cleavage, we used ^1^H–NMR spectroscopy (Figure 3). Polymers were dissolved in CDCl_3_ and irradiated for 2 h. Despite the appearance of new signals at 11 ppm and in the aromatic region (6.5–8 ppm) corresponding to the aldehyde and the nitrosobenzaldehyde derivative, respectively, the cleavage of the side-chain was incomplete after 2 h. Therefore, the irradiation was resumed for one more hour. Surprisingly, the degradation was still incomplete as observed by the presence of the aromatic signals corresponding to the *o*NB linker (Figure 3). The sample was then irradiated overnight and, as shown in Figure 3, the complete disappearance of aromatic signals corresponding to the *o*NB linker was observed. The control sample **PF-TEG** was also irradiated overnight, and no changes were observed (see Appendix A). The difference in the irradiation time evaluated by UV-Vis absorption spectroscopy and ^1^H NMR spectroscopy can be explained by the significant difference in concentration of the samples between the two techniques (~0.1 mg/mL for the UV-Vis absorbance spectroscopy sample compared to ~5 mg/mL for the ^1^H-NMR spectroscopy sample).

### 2.3. Polymer-SWNTs Dispersions and Characterization

Having shown that cleavage of the side-chains is effective on the polymer in solution, we next investigated polymer-SWNTs complexes. Based on our previous work [64], direct dispersion of SWNTs using functionalized polyfluorene led to poorly dispersed samples. Therefore, PF-N_3_ was functionalized post-dispersion. **PF-N_3_** and raw HiPco SWNTs (average tube diameter 0.8–1.2 nm) complexes were prepared following the procedures in the literature [60], Briefly, a mixture of 7.5 mg of **PF-N_3_** and 5 mg of SWNTs in 10 mL of THF was sonicated using a probe sonicator for 1 h. The resulting black suspension was centrifuged at 8346× *g* for 30 min, and the supernatant was carefully removed to isolate the **PF-N_3_-SWNTs** dispersion. The side-chains were then introduced in situ by CuAAC following the procedures in the literature (Figure 2) [60]. The reactions were monitored by IR spectroscopy via the disappearance of the polymer azide stretch at ~2090 cm^−1^ (Appendix A). Once the complete disappearance of the azide stretch was observed, the sample was filtered through a Teflon membrane with 0.2 µm pore diameter and thoroughly rinsed with THF until the filtrate did not fluoresce when excited with a hand-held UV lamp at 365 nm. The resulting thin films were then redispersed in 10 mL of THF.

Characterization using UV−Vis−Near-Infrared (NIR) absorption spectroscopy was next performed (Figure 4). Each SWNTs species present within the polymer-SWNTs sample exhibits its own absorption signals. Three main regions are observed for HiPco SWNTs: two semiconducting regions, S_11_ (830−1600 nm) and S_22_ (600−800 nm), and one metallic region, M_11_ (440−645 nm) [5]. The absorption spectra were normalized to the maximum absorption of the peak at 1140 nm to compare the different SWNTs’ species. **PF-N_3_**-SWNTs and the post-click dispersions (**PF-TEG-SWNTs** and **PF-*o*NB-TEG-SWNTs**) show similar absorption features, suggesting a successful post-functionalization redispersion. As shown in Figure 4, both m- and sc-SWNTs’ species are present within the polymer-SWNTs complexes, suggesting a lack of selectivity for specific SWNTs’ species under the dispersion conditions used.

To further characterize our polymer-SWNTs complexes, Raman spectroscopy was performed. In this technique, laser excitation wavelengths overlap with the Van Hove singularities present in the density of states for specific SWNTs’ species [82]. Therefore, this technique allows examination of both m- and sc-SWNTs’ species present in the polymer-SWNTs sample [83]. Since electronic transitions depend on SWNTs’ diameter and type, multiple excitation wavelengths are needed to achieve full characterization [83]. The polymer-SWNTs samples for Raman spectroscopy were prepared by drop-casting the dispersion onto a silicon wafer, followed by evaporation at RT. A reference sample was prepared by sonicating raw SWNTs in chloroform and depositing on silicon using the same drop-casting method. For HiPco SWNTs, two excitation wavelengths were used: 633 and 785 nm. Using these wavelengths, both m- and sc-SWNTs are separately probed [84]. Figure 5 shows the radial breathing mode (RBM) for **PF-N_3_-SWNTs**, **PF-TEG-SWNTs**, and **PF-oNB-TEG-SWNTs** dispersions. The spectra were normalized to the G-band (~1590 cm^−1^) for comparative analysis. Using the 633 nm excitation wavelength, both m- (175−230 cm^−1^) and sc-SWNTs (240−300 cm^−1^) are in resonance and signals corresponding to both nanotube types are observed (Figure 5A) [85]. We then used the 785 nm excitation wavelength to characterize our samples. sc-SWNTs are primarily in resonance (175–280 cm^−1^) when using this wavelength. When raw HiPco SWNTs are excited at 785 nm, a peak at 265 cm^-1^ corresponding to bundled (10,2) SWNTs is observed [86]. As shown in the spectra (Figure 5B), the intensity of this peak is significantly lower compared to the reference sample of unfunctionalized SWNTs. This indicates that our samples are relatively well dispersed.

### 2.4. Photocleavage Study of the PF-SWNTs Complexes

**PF-TEG-SWNTs** and **PF-*o*NB-TEG-SWNTs** dispersed in THF were transferred into quartz cuvettes and irradiated at 365 nm. The UV-Vis-NIR data were collected every 15 min for the first 2 h, and the final data were collected after overnight irradiation. As shown in Figure 6A, no changes were observed for the control sample. Figure 6B shows that the absorbance of the different SWNTs’ species in the photocleavable sample decreased upon cleavage of the side-chains. As shown in Figure 6B, overnight irradiation caused the dispersion to precipitate, resulting in broad absorption peaks for the nanotubes. This indicates that the loss of side-chains eliminates the steric stabilization of the nanotubes’ dispersion.

### 2.5. Conductivity Measurements

Using UV-Vis-NIR absorption spectroscopy, we observed that the side-chain cleavage is effective and complete after overnight irradiation. We next performed conductivity measurements. **PF-TEG-SWNTs** and **PF-*o*NB-TEG-SWNTs** thin films were prepared by filtering the dispersions through a Teflon membrane with a 0.2 µm pore diameter followed by overnight drying at 75 °C in a vacuum oven. Before irradiation, no conductivity could be detected for either of the samples. The films were then placed into the UV reactor and irradiated at 365 nm overnight. The samples were then washed with hot methanol to eliminate the *o*-nitrosobenzaldehyde derivative produced from the cleavage and placed under vacuum at 75 °C for 1 h (in a vacuum oven). However, no conductivity was detected for either of the samples, probably as a result of the limited light penetration through the black polymer-SWNTs thin films (see the Appendix A for details), thus resulting in limited side-chain cleavage in the solid state.

### 2.6. Characterization of the Polymer-SWNTs Dispersions Post-Irradiation

To verify that the irradiation did not damage the surface of the nanotubes, Raman spectroscopy with an excitation wavelength of 633 nm was performed. Comparing the intensity of the D-band centred at ~1290 cm^−1^ relative to the G-band at ~1590 cm^−1^ provides an indication of the presence of sp^3^ carbon defects [87]. As shown in Figure 7, there is no significant difference between pre-and-post irradiated samples when observing the G and D-band. This suggests that no nanotube defects have been generated upon the irradiation process.

## 3. Materials and Methods

### 3.1. General

Flash chromatography was performed using an Intelliflash280 by AnaLogix. Unless otherwise noted, compounds were monitored using a variable wavelength detector at 254 nm. Solvent amounts used for gradient or isocratic elution were reported in column volumes (CV). Columns were prepared in Biotage^®^ SNAP KP-Sil cartridges using 40–63 µm silica or 25–40 µm silica purchased from Silicycle. ^1^H–NMR and ^13^C-NMR spectra were recorded on Bruker Avance 600 MHz and shift-referenced to the residual solvent resonance. Electrospray MS was performed using a Micromass Quattro triple quadrupole instrument in positive mode. Polymer molecular weights and dispersities were analyzed (relative to polystyrene standards) via GPC using a Waters 2695 Separations Module equipped with a Waters 2414 refractive index detector and a Jordi Fluorinated DVB mixed bed column in series with a Jordi Fluorinated DVB 105 Å pore size column. THF with 2% acetonitrile was used as the eluent at a flow rate of 2.0 mL/min. Sonication was performed using a QSonica Q700 Sonicator equipped with a 13 mm probe at an amplitude of 60 µm and a sonication power of 30 Watts. Centrifugation of the polymer-SWNTs’ samples was performed using a Beckman Coulter Allegra X-22 centrifuge. UV-Vis-NIR absorption spectra were recorded on a Cary 5000 spectrometer in dual beam mode, using matched 10 mm quartz cuvettes. Raman spectra were collected with a Renishaw InVia Laser Raman spectrometer, using two different lasers: a 500 mW HeNe Renishaw laser (633 nm, 1800 L/mm grating); and a 300 mW Renishaw laser (785 nm, 1200 L/mm grating). Photoirradiation of the samples was performed in a home-built UV reactor equipped with two 25W lamps exhibiting emission at 365 nm.

### 3.2. Experimental Procedures

Raw HiPCO SWNTs were purchased from NanoIntegris (batch #HR30-166 and #HR37-033) and used without further purification. All reagents were purchased from commercial chemical suppliers and used as received. Compounds 1, 2, and 3 were prepared according to the procedures in the literature [62]. Synthesis schemes and characterization data are provided in the Appendix A.

#### 3.2.1. Poly(bis(6-bromohexyl)fluorene) (PF-Br) [62]

A Schlenk tube equipped with a stir bar was charged with 2 (0.87 g, 1.34 mmol), 3 (1.00 g, 1.34 mmol), THF (6.7 mL), toluene (6.7 mL), and 3M K_3_PO_4(aq)_ (13.4 mL). The reaction mixture was degassed by three freeze–pump–thaw cycles. The biphasic mixture was frozen under liquid nitrogen, then [(o-tol)_3_P]_2_Pd (14 mg, 20.2 µmol) was added under a positive pressure of nitrogen. The Schlenk tube was evacuated and backfilled three times, and the reaction was vigorously stirred at 60 °C for 2 h 30 min. The phases were allowed to separate, and the organic layer was filtered through a plug of celite and neutral alumina (1:1 composition). The plug was washed with THF, and the filtrate was concentrated in vacuo. The crude polymer was precipitated in MeOH (~200 mL) and filtered to afford PF-Br as a yellow solid (1.16 g, 88%). ^1^H–NMR (600 MHz; CDCl_3_): δ 7.86 (m, 2H), 7.73–7.67 (m, 4H), 3.31–3.28 (m, 4H), 2.15 (m, 4H), 1.71–1.69 (m, 4H), 1.28–1.25 (m, 4H), 1.18 (m, 4H), 0.88 (m, 4H).

#### 3.2.2. Poly(bis(6-azidohexyl)fluorene) (PF-N_3_) [62]

A round bottom flask equipped with a stir bar was charged with PF-Br (1.00 g, 2.03 mmol), NaN_3_ (1.32 g, 20.3 mmol), *^n^*Bu_4_NBr (1.32 g, 4.1 mmol), and THF (200 mL). The reaction mixture was heated to reflux for 24 h. The reaction mixture was filtered through a neutral alumina plug, washed with THF, and precipitated in MeOH (~200 mL) to afford PF-N_3_ (0.73 g, 86%).^1^H–NMR (600 MHz; CDCl_3_): δ 7.87–7.85 (m, 2H), 7.72–7.68 (m, 4H), 3.16–3.14 (m, 4H), 2.15 (m, 4H), 1.46, 1.40 (m, 4H), 1.25–1.20 (m, 8H), 0.88–0.83 (m, 4H).

#### 3.2.3. 2-(2-(2-methoxyethoxy)ethoxy)ethyl 4-methylbenzenesulfonate (4) [88]

A 20 mL vial equipped with a stir bar was charged with triethylene glycol monomethyl ether (1 g, 6.09 mmol), Tosyl chloride (1.05 g, 5.53 mmol) in 6 mL of dichloromethane. Triethylamine (1.23 g, 12.2 mmol) was added dropwise. The solution was stirred for 4 h. The reaction was quenched with water (6 mL) and extracted with DCM (3 × 12 mL). The combined organic layers were washed with brine (3 × 12 mL), dried with MgSO_4_, and concentrated in vacuo. The product was purified by flash chromatography Hex/EtOAc (0% to 75%) to give a colourless oil (1.5 g, 77%). ^1^H–NMR (600 MHz, CDCl_3_): δ 7.82 (d, *J* = 8.2 Hz, 2H), 7.36 (d, *J* = 7.9 Hz, 2H), 4.15 (m, 2H), 3.68 (m, 2H), 3.60 (m, 6H), 3.53 (m, 2H), 3.36 (s, 3H), 2.44 (s, 3H).

#### 3.2.4. 5-(2-(2-(2-methoxyethoxy)ethoxy)ethoxy)-2-nitrobenzaldehyde (5) [89]

A 10 mL vial equipped with a stir bar was charged with 5-Hydroxy-2-nitrobenzaldehyde (0,5 g, 2.9 mmol), compound 4 (0.86 g, 2.7 mmol), and potassium carbonate (0.41 g, 2.9 mmol) in 6 mL of dimethylformamide. The solution was stirred at 50 °C overnight. The solution was quenched with water (6 mL), and the mixture was extracted with EtOAc (3 × 12 mL). The recombined organic layers were washed with water (3 × 12 mL) and brine (3 × 12 mL). The organic layer was dried with MgSO_4_ and concentrated in vacuo. The compound was purified by flash chromatography Hex/EtOAc (0% to 50%) to give a colourless oil (0.78 g, 83%). ^1^H–NMR (600 MHz, CDCl_3_): δ 10.47 (s, 1H), 8.1 (d, *J* = 9.04 Hz, 1H), 7.35 (d, *J* = 2.9 Hz, 1H), 7.19–7.17 (dd, *J* = 9.05, J= 2.9 Hz, 1H), 4.27 (m, 2H), 3.90 (m, 2H), 3.73 (m, 2H), 3.67 (m, 2H), 3.64 (s, 2H), 3.54 (s, 3H), 3.37 (s, 3H).

#### 3.2.5. (5-(2-(2-(2-methoxyethoxy)ethoxy)ethoxy)-2-nitrophenyl)methanol (6) [89]

A 20 mL vial equipped with a stir bar was charged with compound 5 (0.75 g, 2.24 mmol) in 6 mL of dry THF. Sodium borohydride (0.13 g, 3.35 mmol) was added, and the solution was stirred at 0 °C for 1 h. After completion, the mixture was diluted with EtOAc, filtered through a silica plug using EtOAc as eluent, and concentrated in vacuo to give a yellowish solid (0.64 g, 85%). ^1^H–NMR (600 MHz, CDCl_3_): 8.17 (d, *J* = 9.04 Hz, 1H), 7.30 (d, *J* = 2.9 Hz, 1H), 6.90 (dd, *J* = 9.05, J= 2.9 Hz, 1H), 4.98 (s, 2H), 4.47 (m, 2H), 3.89 (m, 2H), 3.73 (m, 2H), 3.67 (m, 2H), 3.64 (m, 2H), 3.54 (m, 2H), 3.37 (s,3H).

#### 3.2.6. 4-(2-(2-(2-methoxyethoxy)ethoxy)ethoxy)-1-nitro-2-((prop-2-yn-1-yloxy)methyl)benzene (oNB-TEG-alkyne)

A 100 mL round bottom flask equipped with a stir bar was charged with compound 6 (0.6 mg, 1.90 mmol) and sodium hydride (54 mg, 2.28 mmol) in 20 mL of dry THF. The solution was stirred at 0 °C for 1 h, and propargyl bromide (0.34 g, 2.85 mmol) was added dropwise. The mixture was stirred overnight at RT, then quenched with water (10 mL), extracted with dichloromethane (3 × 20 mL), washed with water (3 × 30 mL) and brine (3 × 30 mL). The organic layer was dried with MgSO_4_, and concentrated in vacuo. The compound was purified by flash chromatography Hex/EtOAc (0% to 90%) to give a brown oil (0.5 g, 75%). ^1^H–NMR (600 MHz; CDCl_3_): δ 8.16 (d, *J* = 9.1 Hz, 1H), 7.30 (d, *J* = 2.8 Hz, 1H), 6.91 (dd, *J* = 9.1, 2.8 Hz, 1H), 4.99 (d, *J* = 6.5 Hz, 2H), 4.27–4.25 (m, 2H), 3.90–3.88 (m, 2H), 3.74–3.63 (m, 6H), 3.55–3.54 (m, 2H), 3.37 (s, 3H), 2.71 (t, *J* = 6.6 Hz, 1H).^13^C–NMR (151 MHz; CDCl3): δ 163.5, 140.49, 140.34, 127.9, 114.6, 113.9, 71.9, 70.89, 70.70, 70.55, 69.6, 68.2, 62.9, 59.0. ESI-MS: *m*/*z* calculated for C_17_H_23_NO_7_ [M^+^]: 353.15 found [M+Na]^+^: 376.1.

#### 3.2.7. Triethylene glycol methyl propargyl ether (TEG-alkyne) [90]

A round bottom flask equipped with a stir bar was charged with triethylene glycol monomethyl ether (1.00 g, 6.09 mmol) in 60 mL of anhydrous THF at 0 °C. NaH (0.28 g, 6.69 mmol) was added to the solution, and the reaction mixture was stirred at 0 °C. After 1 h, propargyl bromide (80% in toluene, 1.43 g, 7.91 mmol) was added dropwise. The reaction mixture was stirred at RT overnight, quenched with water (30 mL) and extracted with DCM (3 × 30 mL). The organic layers were combined, washed with brine (3 × 25 mL), dried with MgSO_4_ and concentrated in vacuo to afford TEG-alkyne as an orange oil (1.05 g, 85%). ^1^H–NMR (600 MHz; CDCl_3_): δ 4.20 (d, *J* = 2.4 Hz, 2H), 3.71–3.63 (m, 10H), 3.55–3.54 (m, 2H), 3.37 (s, 3H), 2.42 (t, *J* = 2.4 Hz, 1H).

#### 3.2.8. CuAAC Procedure for PF-TEG (P1)

A glass vial was charged with PF-N_3_ (30 mg, 70 µmol) in 7 mL THF and 2.1 equivalents of TEG-alkyne (30 mg, 150 µmol). To the reaction mixture, 4 mg of Cu(OAc) and 10 equivalents of Hünig’s base with respect to Cu(OAc) (40 mg, 0.36 mmol) were added. The reaction mixture was stirred at RT, and reaction progress was monitored by IR spectroscopy for the disappearance of the azide stretch at ~2090 cm^−1^. The solution was filtered through an alumina plug, concentrated in vacuo and redissolved in a minimum of THF. The polymer was precipitated in ∼50 mL of cold MeOH to give a yellow solid (51 mg, 87%). ^1^H–NMR (600 MHz; CDCl_3_): δ 7.90–7.85 (m, 2H), 7.77–7.67 (m, 4H), 7.53–7.48 (m, 2H), 4.69–4.63 (m, 4H), 4.27–4.19 (m, 4H), 3.69–3.64 (m, 18H), 3.58–3.52 (m, 4H), 3.39–3.37 (m, 6H), 2.20–2.09 (m, 4H), 1.79–1.71 (m, 4H), 1.23–1.11 (m, 10H), 0.88–0.77 (m, 4H).

#### 3.2.9. CuAAC Procedure for PF-oNB-TEG (P2)

A glass vial was charged with PF-N_3_ (30 mg, 70 µmol) in 7 mL THF and 2.1 equivalents of *o*NB-TEG-alkyne (50 mg, 150 µmol). To the reaction mixture, 4 mg of Cu(OAc) and 10 equivalents of Hünig’s base with respect to Cu(OAc) (40 mg, 0.36 mmol) were added. The reaction mixture was stirred at RT, and the reaction progress was monitored by IR spectroscopy for the disappearance of the azide stretch at ~2090 cm^−^^1^. The solution was filtered through an alumina plug, concentrated in vacuo and redissolved in a minimum of THF. The polymer was precipitated in ~50 mL of cold MeOH to give a yellow solid (67 mg, 90%). ^1^H–NMR (600 MHz, CDCl_3_): δ 8.11 (m, 2H), 7.83 (m, 2H), 7.68 (m, 4H), 7.48 (m, 2H), 7.32 (m, 2H), 4.95 (m, 4H),4.74 (m, 4H), 4.22(m, 6H), 3.92 (m, 4H), 3.73–3.65 (m, 12H), 3.52 (m, 4), 3.36 (m, 6H), 2.12 (m, 4H), 1.76 (m, 4H), 1.22 (m, 8H), 0.84 (m, 8H).

## 4. Conclusions

Polyfluorene-SWNTs complexes were functionalized with either photocleavable or non-cleavable TEG side-chains. The photocleavage of the side-chains was first studied on the polymer using UV-Vis absorption spectroscopy. As the irradiation time increased, a decrease in the absorbance was observed at 300 nm, while a new peak at 350 nm arose for the cleavable sample. This corresponds to the photoisomerization of the *o*NB linker into the corresponding *o*-nitrosobenzaldehyde. Functionalized polyfluorene-wrapped SWNTs were then prepared, and the photocleavage was studied using UV-Vis-NIR absorption spectroscopy. After overnight irradiation, the cleavage of the side-chains led to the precipitation of the dispersion, resulting in broad absorption peaks. Finally, thin-film samples of the polymer-wrapped SWNTs were prepared and irradiated overnight. Unfortunately, due to the limited ability of the light to penetrate through the thin films, no conductivity could be detected pre-and post-irradiation. However, these results demonstrate the ability to efficiently cleave the side-chains using light without damaging the structure of the nanotubes.

## Data Availability

The data presented in this study are available in the Appendix A.

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
