# Peer review of "Decoration of Polyfluorene-Wrapped Carbon Nanotubes with Photocleavable Side-Chains"

_molecules, 2023, doi:10.3390/molecules28031471_

Round 1

Reviewer 1 Report

The article Decoration of polyfluorene-wrapped carbon nanotubes with photocleavable side-chains is a very interesting study. The authors have improved the dispersion of SWNTs in organic solvents by surface modification of SWNTs with photocleavable side-chains and successfully achieved the photodegradation of side-chains of SWNTs in the solution state. However, there are a few notes that are worth working on, such as:

1. What is the purity of SWNTs used in the work? Their aspect ratio, functional groups on the surface? This is important for later understanding the characteristics of SWNTs derivatives.

2. Section 2.2, line 120, the authors write: a UV reactor (see Supporting Information for details). Where are the details about the UV reactor in the Supporting Information? This is very difficult to find them.

3. According to Figure S8, it can be found that the thin film prepared by the authors are very rough. Under such film preparation process conditions, it is difficult to control the thickness and uniformity. The conductivity of the films prepared by the authors through filtration would be unreliable. If possible, it is best to use a more advanced and controlled process to prepare the film.

4. Section 4, line 414, the authors write: The concept may find potential applications in SWNT-based materials such as OTFTs in which the cleavage of the side-chains could result in higher performance [68]. The work done by the authors is not sufficient to draw conclusions about the material properties and it is not appropriate to cite references in the conclusion of the article.

5. It would be ideal, to give some more details of SWNT-based materials with cleavable side-chains and their device performance from older published works. 

Reviewer 2 Report

The work, titled “Decoration of polyfluorene-wrapped carbon nanotubes using photocleavable cleavable side chains” shows the functionalization of carbon nanotube with polyfluorenes derivatives to improve its dispersibility. Interestingly, the authors performed complex synthesis to provide photocleavable moieties to the polymer to improve the conductivity of the composited after the membrane formation. Although the authors did not accomplish the investigation aim, hence the conductivity of the polyfluorene-SWNT film after its UV-irradiation, this paper gives new knowledge to the field. The manuscript was written in a god English, the method are well described, and the data presented in a very logic way. The introduction provides all the information needed to understand the work. I suggest to the editor the publication of this paper.
